# What Breeds Conspiracy Theories in COVID-19? The Role of Risk Perception in the Belief in COVID-19 Conspiracy

**DOI:** 10.3390/ijerph19095396

**Published:** 2022-04-28

**Authors:** Zhaoxie Zeng, Yi Ding, Yue Zhang, Yongyu Guo

**Affiliations:** School of Psychology, Nanjing Normal University, Nanjing 210097, China; 212301007@njnu.edu.cn (Z.Z.); yiding@njnu.edu.cn (Y.D.); 182301007@njnu.edu.cn (Y.Z.)

**Keywords:** COVID-19, conspiracy theories, risk perception, social governance

## Abstract

Conspiracy theories often emerge during public health crises, and can provide some explanation for the causes behind the crises. However, the prevalence of conspiracy theories also poses a serious threat to public health order and hinders the implementation of disease prevention and control measures. No studies have examined the role of multiple risk perceptions in the formation of beliefs in conspiracy theories from a cognitive perspective in the context of the epidemic. In this cross-sectional study, participants filled in an online survey in order to investigate the relationship between epidemic severity and beliefs in conspiracy theories and the mediating role of risk perception in this relationship. The results showed that COVID-19 epidemic severity positively predicted beliefs in both in- and out-group conspiracy theories. Risk perception mediated the positive relationship between COVID-19 epidemic severity and belief in in-group conspiracy theories. These results suggest that in a major public health crisis event: (1) residents at the epicenter may be more prone to believing in both in- and out-group conspiracy theories; and (2) beliefs in in- and out-group conspiracy theories may have different psychological mechanisms. Therefore, conspiracy theories about public health incidents, such as the COVID-19 pandemic, should be classified and treated by policy stakeholders.

## 1. Introduction

The coronavirus disease 2019 (COVID-19) pandemic is one of the most influential and destructive events faced by human society in modern times. Globally, 428,511,601 confirmed cases and 5,911,081 deaths from COVID-19 have been reported to the World Health Organization as of 5:23 p.m. CET on 24 February 2022 [1]. The pandemic has brought about not only great challenges to global economic development [2] but also fear, uncertainty, and a sense of loss of control owing to its highly threatening nature. Owing to its uncontrollability, the pandemic has also given space for various related conspiracy theories to arise and become popular, such as conspiracy theories related to virus transmission via 5G signal and vaccination [3]. If these conspiracy theories become prevalent, there can be many negative consequences. For instance, researchers have found that beliefs in conspiracy theories related to the COVID-19 vaccine reduced trust in medical science and institutions and led to low objective vaccine knowledge, which in turn predicted high vaccine hesitancy [4]. Another study using a cross-sectional online survey found that participants who believed in conspiracy theories related to COVID-19 reported lower intention to vaccinate than those who disbelieved conspiracy theories [5].

So, why are people more prone to believe in conspiracy theories in the context of the epidemic? Most previous studies were conducted from the perspective of motivation and emotion. Those conducted from the perspective of motivation posit that individuals’ sense of control and sense of meaning will be damaged when the individual is confronted with a crisis. When this happens, conspiracy theories serve the psychological function of compensating for the loss of sense of control and sense of meaning, and, therefore, people tend to believe in them more [6]. Studies conducted from an emotional perspective posit that individuals in crisis experience negative emotions such as anxiety and fear, which are closely related to an increase in beliefs in conspiracy theories [3,7]. Previous studies suggested that it is important to study the causes of conspiracy theories from a cognitive perspective [8]. However, in the context of the epidemic, few studies have explored the causes of conspiracy theories from a cognitive perspective (especially risk perception). To the best of our knowledge, existing research involving risk perception focuses on the health risks posed by the COVID-19 epidemic [9,10,11]. These studies ignore that, in addition to the health risks posed by COVID-19, there is also a threat to people’s quality of life and income. We believe that the perception of such risks is also likely to induce people to embrace conspiracy theories. Therefore, in order to break through the limitations of existing studies, we carried out a nationwide online questionnaire survey among Chinese residents in the early stage of the COVID-19 epidemic. Our survey was based on the following three purposes: first, to examine whether people believe conspiracy theories about the COVID-19 epidemic; second, to examine the role of perceptions of multiple risks in the relationship between epidemic severity and belief in conspiracy theories; finally, to provide relevant referential information for the emergency management of public health disasters.

### 1.1. The Threats of the COVID-19 Pandemic and Belief in Conspiracy Theories

Throughout history, whenever there have been public safety incidents such as terrorist attacks, earthquakes, tsunamis, and infectious diseases, these have been accompanied by various conspiracy theories. Conspiracy beliefs refer to the tendency to interpret certain historical or contemporary events as being implicitly and deliberately premeditated by powerful organizations or individuals to achieve their intended purposes [12]. During the COVID-19 pandemic, the negative effects of conspiracy theories can be seen at the individual, interpersonal, and societal levels. At the individual level, beliefs in conspiracy theories can have adverse effects on one’s mental health and work [13,14]. At the interpersonal level, such beliefs can lead to negative interpersonal consequences, including the endorsement of violent and radical behavior [15]. At the societal level, these beliefs can have a serious negative impact on infectious disease prevention and treatment, such as social distancing defiance and vaccination reluctance [4,15]. Considering these potential adverse effects, a question is begged: what is the source of these conspiracy theories on COVID-19?

According to the existential threat model of conspiracy theories, influential and anxiety-provoking social events inspire beliefs in conspiracy theories [16], and this view is supported by empirical research. Studies have shown that in the hypothetical scenario of the assassination of the president of the United States of America, people were more likely to believe in conspiracy theories when the assassination triggered a war than when it did not trigger a war [17]. This suggests that social events which pose greater threats lead to stronger beliefs in conspiracies than relatively insignificant social events. Accordingly, we believe that the COVID-19 pandemic—as a crisis event that objectively poses a serious threat to one’s life and property, as well as to a sense of certainty in life and control—will constitute an existential threat to individuals, potentially stimulating their beliefs in conspiracy theories. This leads to:

**Hypothesis** **1** **(H1).**
*The more severe the COVID-19 epidemic in an area, the more likely it is for those residing in the area to believe in conspiracy theories about the epidemic.*


### 1.2. The Ripple Effect of Risk Perception: The Threats of the COVID-19 Pandemic and Risk Perception

As aforementioned, the COVID-19 pandemic poses an objective threat to people’s lives and can induce people to believe in conspiracy theories, and we believe that the perception of the risks caused by this threat may be formed through people’s subjective perceptions and judgments of the situation. Risk perception refers to the tendency to use personal subjective intuitive judgments to make a cognitive assessment of the risks of dangerous factors in a situation [18]. In the social sciences, risk perception is defined as beliefs, attitudes, judgments, and emotions about dangers and benefits, and cultural tendencies in a broader sense [19]. For this study, we define risk perception as a subjective judgment of the negative impact of dangerous factors in a situation, including the evaluation of the possibility of dangerous factors occurring and their impact on all aspects of one’s life.

Risk perception influences human self-protection and social behaviors [20]. Some studies have shown that the closer people are to the central area of public crisis events, the stronger their risk perceptions and negative emotions about the event [21,22]. This effect is called the ripple effect, and the term “ripple” is an image metaphor used by the social amplification framework of risk regarding the patterns of the impacts of risk events [21]. Research from different disciplines supports the ripple effect. Lima designed a 5-year longitudinal follow-up study in which 2797 residents living near incinerators were interviewed, showing that risk perception was more acute for residents living closer to the site, who also had a less favorable attitude toward the incinerators [23]. A recent study also found that in the early days of the COVID-19 outbreak, Wuhan citizens (who were closer to the epicenter of the outbreak) had stronger risk perception and cognitive anxiety than those living in other regions of China (farther away from the epicenter), which is consistent with the ripple effect [24]. This leads to:

**Hypothesis** **2** **(H2).**
*Residents of areas with a high COVID-19 epidemic severity level will have stronger risk perceptions than those in areas with a low epidemic severity level.*


### 1.3. The Threats of the COVID-19 Pandemic, Risk Perceptions, and Belief in Conspiracy Theories

Previous studies have explored the relationship of crisis events with beliefs in conspiracy theories and with risk perception. However, as mentioned above, to the best of our knowledge, no study has paid attention to the relationship of these three before. In general, these studies are divided into two categories; studies that examine the causes of conspiracy theories in the context of the COVID-19 epidemic from the perspective of motivation or emotion (e.g., 3, 7), and studies that examine the risk of virus infection and conspiracy theories in the context of the COVID-19 epidemic and that do not consider threat of the epidemic as an independent variable (e.g., 9, 10, 11). Based on these previous studies, we attempted to construct an integrated model to examine the relationship between epidemic severity, risk perception, and beliefs in conspiracy theories. In particular, we have examined the beliefs in conspiracy theories of residents in areas with different levels of COVID-19 epidemic severity. That is, while combining social amplification framework of risk [21] and the existential threat model of conspiracy theories [16], this study explores how epidemic severity (objective factor) predicts people’s beliefs in conspiracy theories through risk perception (subjective psychological factor).

Similar to natural disasters such as earthquakes and tsunamis, the COVID-19 pandemic emerged suddenly, induced huge and widespread economic and social problems, brought forth uncertainty, and its effect has been longitudinally sustained [24,25]. However, the threat that the COVID-19 pandemic poses to life and property, owing to the ability of the causative virus to spread and mutate, is even more elusive than that of other natural disasters; this makes COVID-19 a threat that can cause great pain, anxiety, and uncertainty.

According to the social amplification framework of risk, the interaction of risk events with psychological, social, institutional, and cultural processes affects the public’s risk perception and related behavior [26]. Therefore, we believe that the severer the epidemic in an area, the stronger the risk perception of the local population, showing the “ripple effect” of risk perception. According to the existential threat model of conspiracy theories [16], the uncertainty and anxiety brought about by strong risks related to an epidemic will in turn trigger beliefs in conspiracy theories regarding the event. This leads to:

**Hypothesis** **3** **(H3).**
*Risk perception mediates the relationship between epidemic severity and beliefs in conspiracy theories.*


Specifically, the severer the epidemic in an area, the stronger the risk perception of its residents, and the stronger the likelihood of people to commit to beliefs in conspiracy theories about COVID-19; this may serve to avoid negative psychological feelings evoked by the strong risk perception. The hypothetical model used in this study is illustrated in Figure 1.

## 2. Materials and Methods

### 2.1. Participants and Design

This study was part of an online national survey conducted in China between 31 January and 1 February 2020, at the beginning of the COVID-19 outbreak nationwide. Ethical approval was obtained from the Ethics Committee of Nanjing Normal University (protocol code NNU202110006). The participants were 1524 individuals (705 women and 819 men) aged 18–61 years (mean = 30.49, standard deviation = 8.19). All participants were recruited using Tencent Questionnaire (https://wj.qq.com, accessed on 1 January 2020), a Chinese online platform similar to Amazon’s Mechanical Turk. Further, 36.75% of the participants were from Hubei Province.

Before filling in the questionnaire, all participants read an informed consent form, through which they were informed about the basic content of the study, the study procedure, and their right to withdraw from research at any time of their own volition. Only those who clicked on the “yes” option were allowed to continue to participate in our study. After that, all participants completed measures of risk perception, belief in conspiracy theories, and finally, demographic information.

### 2.2. Measures

#### 2.2.1. COVID-19 Epidemic Severity

To assess COVID-19 epidemic severity, we used a multiple-choice question (the options for this question included 31 provincial-level administrative regions in mainland China), in which participants could choose the provincial-level administrative region they were in. According to the data reported by the National Health Commission of the People’s Republic of China at the beginning and the end of data collection, the number of confirmed COVID-19 cases in Hubei province was much higher than those in the other 30 provincial-level administrative regions, which had a similar number of confirmed cases [27]. Therefore, Hubei province has been defined as severely affected, and non-Hubei provinces as not severely affected.

#### 2.2.2. Risk Perception

To assess risk perception, we used a tool adapted from the Flood Risk Perception Questionnaire [28]. This self-reported tool comprises five items. We measured participants’ risk perceptions about life safety, quality of life, and economic losses, among other factors caused by the epidemic. Specifically, we asked the following questions: (1) “How likely is it that the disease will spread in your community?”; (2) “To what extent does the epidemic threaten your or your family’s life?”; (3) “To what extent has the epidemic affected your quality of life?”; (4) “To what extent will the pandemic cause you economic losses?”; (5) “Are you afraid of the epidemic?”. Each item was scored on a scale from 1 to 7. Scores were calculated by adding the score in each item. Higher scores indicated greater risk perception. The McDonald’s omega of the tool in this study was 0.69.

#### 2.2.3. Belief in Conspiracy Theories

Beliefs in conspiracy theories were assessed by a 4-item tool used in Study 1 of Jolley et al. [29], which measures the degree of belief in conspiracy theories related to the COVID-19 pandemic. In this tool, there are two subscales with two items each: in- and out-group conspiracy theories (two reverse scoring). The items are scored on a 7-point Likert-type scale ranging from 1 (completely wrong) to 7 (completely right). Scores were calculated by adding the score in each item. Higher scores indicated stronger belief in conspiracy theories regarding the COVID-19 pandemic. The correlation coefficient of the two items of belief in in-group conspiracy theories is *r* = 0.56 (*p* < 0.001), and that of the two items of belief in out-group conspiracy theories is *r* = 0.39 (*p* < 0.001).

#### 2.2.4. Demographics

In addition to completing the measurement of the main variables, at the end of the questionnaire, participants were asked to report demographic information, including age, sex, subjective social class, educational background, and income. Table 1 presents demographic information of the participants. In our study, participants came from 27 provincial-level administrative regions in mainland China.

### 2.3. Data Analyses

First, we performed common method bias tests to test that there is no strong method factor contributing to the communality of all dependent variables. Second, descriptive statistical analysis and independent sample *t*-test were conducted to compare the differences in risk perception, beliefs in in-group conspiracy theories, and beliefs in out-group conspiracy theories between Hubei and non-Hubei residents. Finally, we conducted a mediation model to test the role of risk perception in the relationship between epidemic severity and two types of belief in conspiracy theories (belief in in-group conspiracy theories and belief in out-group conspiracy theories). Epidemic severity was the dependent variable, risk perception was the mediator, and belief in conspiracy theories was the independent variable. In addition, although not the main focus of the present study, we also added participants’ subjective social class, income, and educational background in an additional model to test the potential influences of these variables on participants’ beliefs in conspiracy theories.

## 3. Results

### 3.1. Testing and Controlling for Common Method Biases

As suggested by previous researchers [30], we controlled for common method biases by using anonymized questionnaires, reverse scoring in some items, and adding items to check for attention (three such items were included in the national questionnaire project of this study). After data collection completion, Harman’s one-factor test was used to test for common method biases; there were three factors with eigenvalues greater than one, with the first factor explaining 24.4% of the variance. This was lower than the critical standard of 40% [30].

### 3.2. Preliminary Analyses

Descriptive statistics and independent sample *t*-tests were performed using SPSS, version 26 (International Business Machines Corporation, IBM, Armonk, NY, USA). Independent sample *t*-tests showed that Hubei residents (i.e., those who were at the center of the epidemic) had significantly stronger risk perception (*p* < 0.001, Cohen’s d = 0.27) and were more committed to beliefs in in-group (*p* < 0.001, Cohen’s d = 0.21) and out-group conspiracy theories (*p* = 0.002, Cohen’s d = 0.16) than non-Hubei residents. The results are shown in Table 2.

A one-way repeated measures ANOVA was subsequently conducted using region (Hubei vs. other regions) as a between-subjects variable and belief in conspiracy theories (in-group conspiracy theories vs. out-group conspiracy theories) as a within-subjects variable. The results showed a significant main effect of region, F (1, 1522) = 22.50, *p* < 0.001, η^2^_p_ = 0.015, and belief in conspiracy theories was significantly stronger among residents of Hubei than non-Hubei regions. The main effect of conspiracy theories type was significant, F (1, 1522) = 59.66, *p* < 0.001, η^2^_p_ = 0.038; belief in in-group conspiracy theories was significantly stronger than belief in out-group conspiracy theories. As shown in Table 3, the results of post hoc multiple comparisons using Bonferroni’s method found no significant difference between in-group conspiracy theories of non-Hubei residents and out-group conspiracy theories of Hubei residents in all pairs of comparisons, and significant differences were observed in the remaining comparisons (*p* < 0.01).

### 3.3. Mediation Analysis

#### Testing the Mediating Effect of Risk Perception on the Relationship between Beliefs in In-Group Conspiracy Theories and COVID-19 Epidemic Severity

Following past recommendations [31], data were processed and analyzed by SPSS, version 26 (International Business Machines Corporation, IBM, Armonk, NY, USA), and SPSS Amos, version 22 (International Business Machines Corporation, IBM, Armonk, NY, USA), using the bias-corrected nonparametric percentile bootstrap test with 5000 resamples. In the analysis, the non-Hubei sample was coded as 1 and the Hubei sample was coded as 2.

The results showed (as shown in Table 4 and Figure 2) that the direct effect of epidemic severity on risk perception was 0.13; the total effect of epidemic severity on in-group conspiracy beliefs was 0.10 and the indirect effect was 0.021 (95% CI, 0.012 to 0.033); the total effect of epidemic severity on out-group conspiracy beliefs was 0.08, the direct effect was 0.08, and the indirect effect was 0.005 (95% CI, −0.002 to 0.014). These results suggest that risk perception partially mediates the relationship between COVID-19 epidemic severity and beliefs in in-group conspiracy theories, but does not play a mediating role when the subjects believe in out-group conspiracy theories.

Previous studies have shown that subjective social class, income, and educational background are significantly and negatively correlated with beliefs in conspiracies [32,33]. In our findings, even while controlling for subjective social class, income, and educational background, the analyzed mediating effect still showed the value of 0.06 (95% CI = 0.03, 0.10).

To ensure the robustness of the results, we also asked participants to report on the situation of the COVID-19 epidemic in their regions, including seven scenarios: (1) high-incidence area with residents in medical isolation; (2) high-incidence area with residents not in medical isolation; (3) the area was recently a high-incidence area, but the situation has improved; (4) there is a small number of cases in the area, but the impact is not large; (5) there is no COVID-19 epidemic in the area; (6) there is no COVID-19 epidemic in infectious disease hospitals in the region; and (7) unclear. The latter two scenarios (with 5 and 40 participants, respectively, having reported them) were excluded from the analysis because COVID-19 epidemic severity could not be graded. The results of the main analyses were consistent with those of the analysis conducted while using region as an independent variable.

In short, COVID-19 epidemic severity positively predicted beliefs in conspiracy theories and risk perceptions, verifying H1 and H2. Regarding mediating effects, risk perception played a partial mediating role in the relationship of COVID-19 epidemic severity with beliefs in in-group conspiracy theories, but not with beliefs in out-group conspiracy theories. These findings partially supported H3.

## 4. Discussion

### 4.1. Contribution and Implications

This study shows that, first, residents in Hubei, where the COVID-19 epidemic was severer at the time of this study, were more likely to believe in conspiracy theories about COVID-19 than those in other areas where the epidemic was less severe. Upon the onset of the COVID-19 pandemic, it became harder for people’s psychological needs regarding a sense of certainty and control to be met, and people became worried about and afraid of the future. Specifically, research shows that crisis events indeed lead people to start seeking answers to important questions about work, study, and more in the months and years after crisis onset, all while often facing an extremely complex, and sometimes even contradictory, information environment [8]. The present study supports the notion that, when people seek information or solutions to conundrums that are both difficult and uncertain, conspiracy theories often serve as easy and available explanations [34]. Second, study participants reported on their own risk perceptions regarding COVID-19, which at the time of the study was still considered an epidemic, including the risks pertaining to the possibility of the spread of the virus and its impact on their own economic status, safety, and quality of life. The results showed that people’s risk perception of COVID-19 epidemic severity was in line with the ripple effect framework, and this is consistent with the findings in previous disaster studies [24]. Third, risk perception was shown to play a partial mediating role in the relationship of COVID-19 epidemic severity with beliefs in in-group conspiracy theories, but not with beliefs in out-group conspiracy theories. We believe that Hubei residents would have stronger risk perceptions, in turn leading to higher beliefs in in-group conspiracy theories. This is in line with the perspective of the society amplification framework of risk [21] and the existential threat model of conspiracy theories [16]. In the early stages of the COVID-19 epidemic, owing to the suddenness of the outbreak and the lack of a smooth transmission of information on the epidemic, the response of relevant departments in the Hubei Province government lagged to a certain extent [35]. These characteristics of the situation at the time implied that the perceived risk was more likely to derive from the in-group rather than the out-group, thus leading to different indirect effects of mediated models.

This study makes several contributions. First, it can help us understand the role of risk perception in shaping conspiracy beliefs in the context of COVID-19. We operationalized by examining the psychological mechanisms of the formation of such beliefs in areas across China with different levels of COVID-19 epidemic severity and using a nationally representative sample. One of our main findings is that Hubei residents, who were in the epicenter of the outbreak of COVID-19, were more likely to believe not only in in-group but also in out-group conspiracy theories. This is in line with the prevailing view in conspiracy theory research that belief in one conspiracy theory reinforces beliefs in other conspiracy theories [36]. Our study also shows that there may be differences in the intrinsic psychological mechanisms of the beliefs in these two types of conspiracy theories, providing a potential research pathway to be explored in the future.

Second, this study validates the “ripple effect” regarding risk perception within the context of public health emergencies. In particular, risk perception was significantly higher in Hubei than in non-Hubei residents, and this finding is consistent with the results of a recent study [24].

Third, this study found that residents in areas severely affected by the COVID-19 epidemic believed in both in-group and out-group conspiracy theories; this finding can serve as reference information for emergency management in the context of COVID-19. The spread of conspiracy theories may hinder the implementation of epidemic prevention and control policies such as, for example, vaccination [37,38]. In addition, conspiracy theories themselves can become sources of threat to people’s lives by fueling the spread of a conspiracy mentality [16] and triggering a “psychological epidemic.” Therefore, while delivering emergency medical services, relevant departments related to psychological care should pay attention to the psychological changes of populations in areas at the center of epidemic outbreaks and help prevent a “psychological epidemic”.

### 4.2. Limitations and Prospects

Before closing, we will briefly outline several limitations and prospects for future research. First, the current study adopted a cross-sectional design. The relationship between the research variables was investigated through an online questionnaire, and the causal relationship between the variables cannot be reliably inferred. Future research can make up for this limitation through the following two aspects. One is to adopt a laboratory design, for example, by presenting priming materials to manipulate participants’ judgments about their own dangerous situations, and then measuring their risk perceptions and beliefs in conspiracy theories. The second is to adopt a longitudinal tracking design to improve the grasp of causal inferences to a certain extent. In addition, compared with the experimental manipulation method, the longitudinal tracking design has higher ecological validity and improves the generalizability of the conclusions.

Second, our results show that risk perception mediates the relationship between epidemic severity and belief in in-group conspiracy theories but there is no mediating effect of risk perception in the relationship between epidemic severity and belief in out-group conspiracy theories. Future studies should further analyze the psychological mechanisms underlying the relationship between epidemic severity and different types of beliefs in conspiracy theories, such as examining the role of system justification [39,40,41] and collective narcissism [42,43,44]. Finally, the spread of COVID-19 is highly random and dynamic, and an outbreak may occur in different regions at different times. However, this study only focused on the static relationship between the two. Therefore, future research can study the dynamic changes of beliefs in conspiracy theories according to the development and changes of the epidemic and improve our understanding of the causes of conspiracy theories in the context of epidemics.

## 5. Conclusions

The results of this research suggest that compared with residents in areas where the epidemic was less severe or non-existent, residents in Hubei (where the epidemic was severer) believed more in conspiracy theories, both in their in-group and out-group formats. We also found that people in areas where the epidemic was severe had a stronger risk perception, showing a trend toward the “ripple effect”. Further, we showed the mediating role of risk perception in the relationship between COVID-19 pandemic severity and beliefs in in-group conspiracy theories, indicating that residents in areas where the COVID-19 epidemic was severe may have a stronger risk perception and may prefer to support in-group conspiracy theories. Still, this effect was not observed in the relationship between COVID-19 epidemic severity and beliefs in out-group conspiracy theories.

## Figures and Tables

**Figure 1 ijerph-19-05396-f001:**
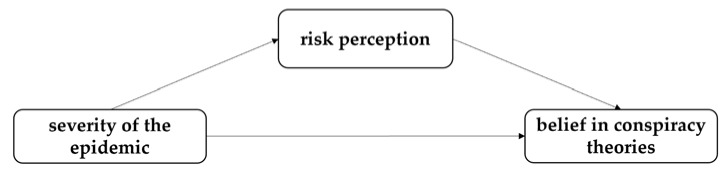
Hypothesized model.

**Figure 2 ijerph-19-05396-f002:**
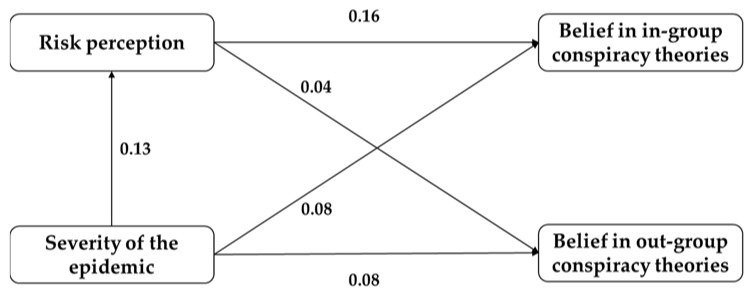
Path model with standardized path coefficients.

**Table 1 ijerph-19-05396-t001:** Demographic information of the participants (*N* = 1524).

Variable	Categories	Frequency	Percentage (%)
Age	18–24	375	24.6%
	25–30	562	36.9%
	31–40	376	24.7%
	41–50	166	10.9%
	51–61	45	2.9%
Gender	female	705	46.3%
	male	819	53.7%
Educational background	Primary school or less	10	0.7%
	Middle school graduate	45	3%
	High school graduate or equivalent education completed	121	7.9%
	Junior college graduate	437	28.7%
	College graduate	748	49%
	Postgraduate degree	163	10.7%
Average personal monthly income in Chinese yuan (i.e., CNY)	CNY < 1000	100	6.6%
	CNY 1000–2000	101	6.6%
	CNY 2000–3000	152	10%
	CNY 3000–5000	379	24.9%
	CNY 5000–8000	355	23.3%
	CNY 8000–12,000	251	16.5%
	CNY 12,000–15,000	77	5.1%
	CNY 15,000–20,000	47	3.1%
	CNY > 20,000	62	4.1%
Subjective social class	1	73	4.8%
	2	98	6.4%
	3	295	19.4%
	4	291	19.1%
	5	389	25.5%
	6	250	16.4%
	7	95	6.4%
	8	27	1.8%
	9	3	0.2%
	10	3	0.2%

**Table 2 ijerph-19-05396-t002:** Descriptive statistics and independent sample *t*-tests of risk perceptions and conspiracy theories among residents of different regions (*N* = 1524).

Variables	Regional Divisions	M ± SD	*t*	*p*	Cohen’s d
Risk perception	Non-Hubei Province	4.35 ± 1.02	−5.14	<0.001	0.27
	Hubei Province	4.64 ± 1.12			
In-group conspiracy theories	Non-Hubei Province	3.07 ± 1.39	−4.00	<0.001	0.21
	Hubei Province	3.37 ± 1.44			
Out-group conspiracy theories	Non-Hubei Province	2.67 ± 1.58	−3.13	0.002	0.16
	Hubei Province	2.94 ± 1.71			

**Table 3 ijerph-19-05396-t003:** Post hoc comparisons of region x belief in conspiracy theories (*N* = 1524).

Comparison					Mean Difference	SE	df	*t*	*p*
Conspiracy Theories Type	Region		Conspiracy Theories Type	Region					
in-group	other regions	-	in-group	Hubei	−0.30	0.08	2994	−3.70	0.001
		-	out-group	other regions	0.40	0.06	1522	6.20	< 0.001
		-	out-group	Hubei	0.12	0.08	2994	1.53	0.752
in-group	Hubei	-	out-group	other regions	0.70	0.08	2994	8.66	<0 .001
		-	out-group	Hubei	0.42	0.09	1522	4.99	< 0.001
out-group	other regions	-	out-group	Hubei	−0.27	0.08	2994	−3.42	0.004

**Table 4 ijerph-19-05396-t004:** Analysis of total, direct, and indirect effects regarding the studied relationships (*N* = 1524).

Outcome Variables	Predictive Variables	Direct Effects	Indirect Effects	Total Effect
Risk perception	Severity of the epidemic	0.13		0.13
Belief in in-group conspiracy theories		0.08	0.021	0.10
Belief in out-group conspiracy theories		0.08	0.005	0.08
Belief in in-group conspiracy theories	Risk perception	0.16		
Belief in out-group conspiracy theories		0.04		

Note: the effect values reported in Table 4 are standardized coefficients.

## Data Availability

Data are available from the corresponding author upon reasonable request.

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
