# Peer review of "What Breeds Conspiracy Theories in COVID-19? The Role of Risk Perception in the Belief in COVID-19 Conspiracy"

_ijerph, 2022, doi:10.3390/ijerph19095396_

Round 1

Reviewer 1 Report

The authors present an informative study contributing to evidence that severity of a crises is related with increased beliefs in conspiracy theories. Moreover, this relation seems to be partially mediated by risk perception.

The topic is timely and of relevance, the study appears sound and convincing, and the manuscript is well written. The major issues with this study may be its cross-sectional design (i.e., causal direction cannot be tested) and the use of relatively short screening tools comprising very few questions only. Although the latter appear to be sufficiently internally consistent, it provokes the question if the scales are sufficiently broad to capture the theoretical extensions of constructs like conspiracy ideation. As the least, more information is required what exactly is captured by the items.

The test of hypotheses should be ideally conducted as a simultaneously estimated path model comprising all relevant variables. A plot of a corresponding path model with standardized path coefficients would help the reader pick up easily the most important results.

Apart from that I found the paper generally convincing. Only few relatively minor recommendations are listed below in chronological order.

Minor issues

- For consistency, capitalize all family names or none.

- L. 90 (and other places): I suggest replacing "severer" by "more severe" (arguably, a personal preference).

- L. 127/ 129: "i.e." could be replaced by "e.g." as there are quite a few papers offering evidence in this direction. 

- Figure 1: H3 is placed somewhat misleading; possibly remove all hypotheses from the Figure and refer to the text instead?

- L. 161 ff.: As to the sample: (1) Were these data collected as part of a larger study? (2) There were relatively many participants from Hubei region relative to other regions. Naturally, these may have been more interested in Covid-related issues (including conspiracy ideation) than people from other regions. In turn, this could have contributed to a bias in the sample. 

- L. 197: Instead of or additionally to Cronbach's alpha, McDonalds's omega could be reported. 

- L. 206/l. 207: Insert "r=" when reporting correlations.

- L. 213: "The sample of this study may represent the general population of China": At least the sample is large and comprises participants from different regions. If the authors want to claim representativeness, they should explicitly test the distribution of demographic variables with national statistics.

- Table 1: How was social class defined?

- L. 217: "We performed common method bias tests to ensure that there was no common method bias in the current study.": Maybe "…to test that there is no strong method factor contributing to the communality of all dependent variables"? (A multi-method approach would be required to disentangle method variance from substantial variance.)

- Table 2: Effect sizes of the differences should be provided. Possibly, add a 2x2 test to test if ingroup and outgroup conspiracy beliefs are differentially affected by residence (Hubei vs. other regions).

- L. 255 ff: I think the best way to test all hypotheses simultaneously would be to conduct a path analysis with both conspiracy beliefs as correlated dependent variables. The direct and indirect effects can be computed analogously by estimating product terms. Accordingly, the paper would profit from reporting the core findings in form of a path diagram with standardized path coefficients.

- L. 269: "while controlling for subjective social class, income, and educational background": How were these covariates controlled for in the analyses (i.e., removed from all variables of only from the DVs; all simultaneously or one-by-one)?

- Discussion: The difference in the mediation effect between ingroup and outgroup conspiracy beliefs deserves a quick discussion.

Reviewer 2 Report

I have read with interest this paper due to the theme is important but I considered that there is an important mistake in the desing: Have the authors used a validate instrument? Why, due the theme, have not used the qualitative methods? For me, it is very important.

Round 2

Reviewer 2 Report

I undestand the reasons of the authors but I don´t agree with them. Nevertheless, the paper has been improved and it can be published.